# Continuous Neural Algorithmic Planners

**Yu He**
University of Cambridge
yh441@cam.ac.uk

**Petar Veličković**
DeepMind
petarv@google.com

**Pietro Liò**
University of Cambridge
pl219@cam.ac.uk

**Andreea Deac**
Mila, Université de Montréal
deacandr@mila.quebec

## Abstract

Neural algorithmic reasoning studies the problem of learning algorithms with neural networks, especially using graph architectures. A recent proposal, XLVIN, reaps the benefits of using a graph neural network that simulates the value iteration algorithm in deep reinforcement learning agents. It allows model-free planning without access to privileged information about the environment, which is usually unavailable. However, XLVIN only supports discrete action spaces, and is hence nontrivially applicable to most tasks of real-world interest. We expand XLVIN to continuous action spaces by discretization, and evaluate several selective expansion policies to deal with the large planning graphs. Our proposal, CNAP, demonstrates how neural algorithmic reasoning can make a measurable impact in higher-dimensional continuous control settings, such as MuJoCo, bringing gains in low-data settings and outperforming model-free baselines.

## 1 Introduction

Graph Neural Networks (GNNs) [1–3] have recently attracted attention in performing algorithmic reasoning tasks [4]. Due to the close algorithmic alignment, GNNs were shown to bring better sample efficiency and generalization ability [5, 6] when learning algorithms such as shortest-path and spanning-tree. There have been a number of other successful applications, covering a range of problems such as bipartite matching [7], min-cut problem [8], and Travelling Salesman Problem [9].

We look at the application of using a GNN that simulates value iteration algorithm [10] in Reinforcement Learning (RL) problems. Value iteration [11] is a dynamic programming algorithm that guarantees to provide an optimal solution, but it is inhibited by the requirement of tabulated inputs. Earlier works [12–16] introduced value iteration as an inductive bias to facilitate RL agents to perform implicit planning, but were found to suffer from an algorithmic bottleneck [17]. Conversely, eXecuted Latent Value Iteration Net (XLVIN) [17] was proposed to leverage a value-iteration-behaving GNN [10] by adopting the neural algorithmic framework [4]. XLVIN is able to learn under a low-data regime, tackling the algorithmic bottleneck suffered by other implicit planners.

So far, XLVIN only applies to environments with *small, discrete* action spaces. The difficulty of a *continuous* action space comes from the infinite pool of action spaces. Furthermore, XLVIN builds a planning graph, over which the pre-trained GNN can simulate value iteration. The construction of the planning graph requires an enumeration of the action space – starting from the current state and expanding for a number of hops equal to the planning horizon. The graph size quickly explodes as the dimensionality of the action space increases, preventing XLVIN from more *complex* problems.

Nevertheless, continuous control is of significant importance, as most simulation or robotics control tasks [18] have continuous action spaces by design. High complexity also naturally arises as the problem moves towards more powerful real-world domains. To extend such an agent powered by neural algorithmic reasoning to *complex, continuous* control problems, we propose **Continuous Neural**

Y. He et al., Continuous Neural Algorithmic Planners. *Proceedings of the First Learning on Graphs Conference (LoG 2022)*, PMLR 198, Virtual Event, December 9–12, 2022.

**Algorithmic Planner (CNAP)**. It generalizes XLVIN to continuous action spaces by discretizing them through binning. Moreover, CNAP handles the large planning graph by following a sampling policy that carefully selects actions during the neighbor expansion stage.

Beyond extending the XLVIN model, our work also opens up the discussion on handling large state spaces in neural algorithmic reasoning. The main motivation for using GNNs to learn algorithms comes from the benefit of breaking the input constraints of classical algorithms and handling raw input data directly with neural networks. Therefore, the large state space problem goes beyond the RL context as we move to apply GNNs with algorithmic reasoning power to other tasks.

In this paper, we confirm the feasibility of CNAP on a continuous relaxation of a classical low-dimensional control task, where we can still fully expand all of the binned actions after discretization. Then, we apply CNAP to general MuJoCo [19] environments with complex continuous dynamics, where expanding the planning graph by taking all actions is impossible. By expanding the application scope from simple discrete control to complex continuous control, we show that such an intelligent agent with algorithmic reasoning power can be applied to tasks with more real-world interests.

## 2 Background

### 2.1 Markov Decision Process (MDP)

A reinforcement learning problem can be formally described using the MDP framework. At each time step $t \in \{0, 1, ..., T\}$, the agent performs an action $a_t \in \mathcal{A}$ given the current state $s_t \in \mathcal{S}$. This spawns a transition into a new state $s_{t+1} \in \mathcal{S}$ according to the transition probability $p(s_{t+1}|s_t, a_t)$, and produces a reward $r_t = r(s_t, a_t)$. A policy $\pi(a_t|s_t)$ guides an agent by specifying the probability of choosing an action $a_t$ given a state $s_t$. The trajectory $\tau$ is the sequence of actions and states the agents took $(s_0, a_0, ..., s_T, a_T)$. We define the infinite horizon discounted return as $R(\tau) = \sum_{t=0}^{\infty} \gamma^t r_t$, where $\gamma \in [0, 1]$ is the discount factor. The goal of an agent is to maximize the overall return by finding the optimal policy $\pi^* = \text{argmax}_\pi \mathbb{E}_{\tau \sim \pi}[R(\tau)]$. We can measure the desirability of a state $s$ using the state-value function $V^*(s) = \mathbb{E}_{\tau \sim \pi^*}[R(\tau)|s_t = s]$.

### 2.2 Value Iteration

Value iteration is a dynamic programming algorithm that computes the optimal policy's value function given a tabulated MDP that perfectly describes the environment. It randomly initializes $V^*(s)$ and iteratively updates the value function of each state $s$ using the Bellman optimality equation [11]:

$$V_{i+1}^*(s) = \max_{a \in \mathcal{A}} \{r(s, a) + \gamma \sum_{s' \in \mathcal{S}} p(s'|s, a) V_t^*(s')\} \tag{1}$$

and we can extract the optimal policy using:

$$\pi^*(s) = \text{argmax}_{a \in \mathcal{A}} \{r(s, a) + \gamma \sum_{s' \in \mathcal{S}} p(s'|s, a) V^*(s')\} \tag{2}$$

### 2.3 Message-Passing GNN (MPNN)

Graph Neural Networks (GNNs) generalize traditional deep learning techniques onto graph-structured data [20][21]. A message-passing GNN [2] iteratively updates its node feature $\vec{h}_s$ by aggregating messages from its neighboring nodes. At each timestep $t$, a message can be computed between each connected pair of nodes via a message function $M(\vec{h}_s^t, \vec{h}_{s'}^t, \vec{e}_{s' \to s})$, where $\vec{e}_{s' \to s}$ is the edge feature. A node receives messages from all its connected neighbors $\mathcal{N}(s)$ and aggregates them via a permutation-invariant operator $\bigoplus$ that produces the same output regardless of the spatial permutation of the inputs. The aggregated message $\vec{m}_s^t$ of a node $s$ can be formulated as:

$$\vec{m}_s^t = \bigoplus_{s' \in \mathcal{N}(s)} M(\vec{h}_s^t, \vec{h}_{s'}^t, \vec{e}_{s' \to s}) \tag{3}$$

The node feature $\vec{h}_s^t$ is then transformed via an update function $U$:

$$\vec{h}_s^{t+1} = U(\vec{h}_s^t, \vec{m}_s^t) \tag{4}$$

## 2.4 Neural Algorithmic Reasoning

A dynamic programming (DP) algorithm breaks down the problem into smaller sub-problems, and recursively computes the optimal solutions. DP algorithm has a general form:

$$\text{Answer}[k + 1][i] = \text{DP-Update}(\{\text{Answer}[k][j]\}, j = 1...n) \tag{5}$$

We can interpret GNN process as DP algorithms [5] by aligning GNN's message-passing step with DP's update step. Let $k$ be the current iteration, and $i$ be the node. A GNN node aggregates messages $\vec{m}_i^k$ from its neighbors and updates its node representation to $\vec{h}_i^{k+1}$. Similarly, a DP algorithm aggregates answers from sub-problems Answer$[k][j]$, then updates its own Answer$[k + 1][i]$. The alignment can thus be seen from mapping GNN's node representation $\vec{h}_i^k$ to Answer$[k][i]$, and GNN's aggregation function to DP-Update.

Previous work [5] proved that GNN could simulate DP algorithms with better sample efficiency and generalization due to their close alignment. Furthermore, [6] showed that learning the individual steps of graph algorithms using GNNs brings generalization benefits. Results from [6][10] also showed that MPNN with max aggregator had the best performance among a range of GNN models.

# 3 Related Work

## 3.1 Continuous action space

A common technique for dealing with continuous control problems is to discretize the action space, converting them into discrete control problems. However, discretization leads to an explosion in action space. [22] proposed to use a policy with factorized distribution across action dimensions, and proved it effective on high-dimensional complex tasks with on-policy optimization algorithms. Moreover, the explosion in action space also requires sampling when constructing the planning graph. Sampled MuZero [23] extended MuZero [24] with a sample-based policy based on parameter reuse for policy iteration algorithms. Our work differs in the way that the sampling policy should be aware of the algorithmic reasoning context. The actions sampled would directly participate in the Bellman optimality equation (Eq.1), and ideally should allow the pre-trained GNN to simulate value iteration optimally in each iteration step.

## 3.2 Large-scale graphs

Sampling modules [25] are introduced into GNN architectures to deal with large-scale graphs as a result of neighbor explosion from stacking multiple layers. The unrolling process to construct a planning graph requires node-level sampling. Previous work GraphSAGE [26] introduces a fixed size of node expansion procedure into GCN [1]. This is followed by PinSage [27], which uses a random-walk-based GCN to perform importance-based sampling. However, our work looks at sampling for implicit planning, where the importance of each node in sampling is more difficult to understand due to the lack of an exact description of the environment dynamics. Furthermore, sampling in a multi-dimensional action space also requires more careful thinking in the decision-making process.

# 4 Architecture

Our architecture uses XLVIN as a starting point, which we introduce first. This is followed by a discussion of the challenges that arise from extending XLVIN to a continuous action space and the approaches we proposed to address them.

## 4.1 XLVIN modules

Given the observation space $S$ and the action space $\mathcal{A}$, we let the dimension of state embeddings in the latent space be $k$. The XLVIN architecture can be broken down into four modules:

**Encoder** ($z : S \to \mathbb{R}^k$): A 3-layer MLP which encodes the raw observation from the environment $s \in S$, to a state embedding $\vec{h}_s = z(s)$ in the latent space.

**Transition** ($T : \mathbb{R}^k \times \mathcal{A} \to \mathbb{R}^k$): A 3-layer MLP with layer norm taken before the last layer that takes two inputs: the state embedding of an observation $z(s) \in \mathbb{R}^k$, and an action $a \in \mathcal{A}$. It predicts the

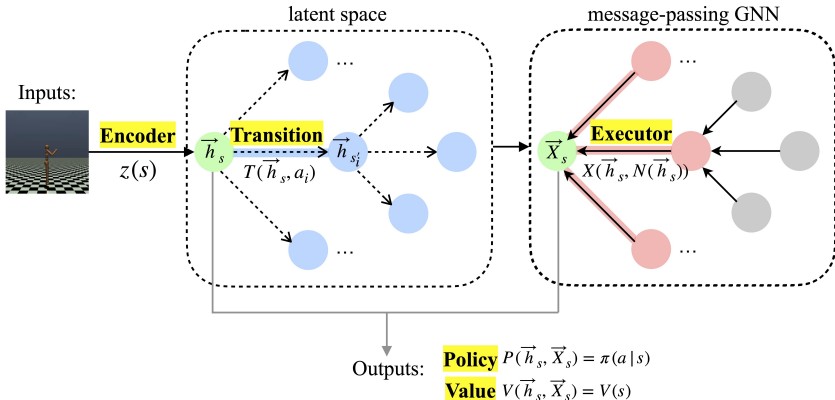

**Figure 1:** XLVIN modules

next state embedding $z(s') \in \mathbb{R}$, where $s'$ is the next state transitioned into when the agent performed an action $a$ under current state $s$.

**Executor** ($X : \mathbb{R}^k \times \mathbb{R}^{|\mathcal{A}| \times k} \to \mathbb{R}^k$): A message-passing GNN pre-trained to simulate each individual step of the value iteration algorithm following the set-up in [10]. Given the current state embedding $\vec{h}_s$, a graph is constructed by enumerating all possible actions $a \in \mathcal{A}$ as edges to expand, and then using the Transition module to predict the next state embeddings as neighbors $\mathcal{N}(\vec{h}_s)$. Finally, the Executor output is an updated state embedding $\vec{\mathcal{X}}_s = X(\vec{h}_s, \mathcal{N}(\vec{h}_s))$.

**Policy and Value** ($P : \mathbb{R}^k \times \mathbb{R}^k \to [0,1]^{|\mathcal{A}|}$ and $V : \mathbb{R}^k \times \mathbb{R}^k \to \mathbb{R}$): The Policy module is a linear layer that takes the outputs from the Encoder and Executor, i.e. the state embedding $\vec{h}_s$ and the updated state embedding $\vec{\mathcal{X}}_s$, and produces a categorical distribution corresponding to the estimated policy, $P(\vec{h}_s, \vec{\mathcal{X}}_s)$. The Tail module is also a linear layer that takes the same inputs and produces the estimated state-value function, $V(\vec{h}_s, \vec{\mathcal{X}}_s)$.

The training procedure follows the XLVIN paper [17], and Proximal Policy Optimization (PPO) [28] is used to train the model, apart from the Executor. We use the PPO implementation and hyperparameters by [29]. The Executor is pre-trained as shown in [10] and directly plugged in.

## 4.2 Limitations of XLVIN

**Discrete control**: XLVIN agents can only choose an action from a discrete set, such as pushing left or right, but not from a continuous range, such as pushing with a magnitude in the range of [0, 1].

**Small action space**: XLVIN is limited by a small size of the action space, while complex control problems come with large action spaces. More importantly, as dimensionality increases, the action space experiences an explosion in size. Take an example of a 3-dimensional robotic dog that operates its 6 joints simultaneously. If we discretize the action space into 10 bins in each dimension, this leads to an explosion in action space to a size of $10^{6^3}$, which exceeds the average computation capacity.

## 4.3 Discretization of the continuous action space

Assume the continuous action space $\mathcal{A}$ has $D$ dimensions, we discretize each dimension $\mathcal{A}_i$ into $N$ evenly spaced actions $\{a_i^1, a_i^2, ..., a_i^N\}$ via binning. However, the discretization of a multi-dimensional continuous action space leads to a combinatorial explosion in action space size. There are two architectural bottlenecks in XLVIN that require an enumeration of all actions, limiting its ability to handle such a large action space.

**The first bottleneck:** The policy layer computes the probability of choosing each action given the current state, resulting in a layer dimension of $N^D$. We chose to use a factorized joint policy in Section 4.4, which reduces the dimension down to $N * D$.

**The second bottleneck:** A planning graph is constructed for the pre-trained GNN to simulate value iteration behavior. Given the current state as a node, we enumerate the action space for neighbor expansion, leading to $N^D$ edges per node. We proposed to use a neighbor sampling policy in Section 4.5 that samples a much smaller number of actions $K \ll N^D$ during neighbor expansion.

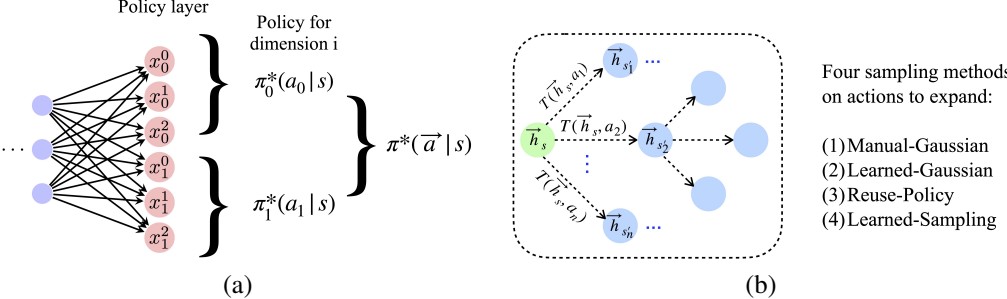

**Figure 2:** (a) Factorized joint policy on an action space with dimension of two. (b) Neighbor sampling methods when constructing the planning graph in Executor.

## 4.4 Factorized joint policy

A naive policy layer $\pi^* = p(\vec{a}|s)$ produces a categorical distribution with $N^D$ logits. To overcome the first bottleneck, we follow a factorized joint policy proposed in [22]:

$$\pi^*(\vec{a}|s) = \prod_{i=1}^{D} \pi_i^*(a_i|s) \tag{6}$$

As illustrated in Figure 2(a), a factorized joint policy $P(\vec{h}_s, \vec{\mathcal{X}}_s)$ is a linear layer with an output dimension of $N * D$. Each policy $\pi_i^*(a_i|s)$ indicates the probability of choosing an action $a_i \in \mathcal{A}_i$ in the $i^{th}$ dimension, where $|\mathcal{A}_i| = N$. This reduces the exponential explosion of action space due to increased dimensionality down to linear. Note there is a trade-off in the choice of $N$, as a larger number of action bins retains more information from the continuous action space, but it also implies larger graphs and hence computation costs. We provide an ablation study in evaluation on the impact of this choice.

## 4.5 Neighbor sampling methods

As illustrated in Figure 2(b), the second bottleneck occurs when constructing a graph to execute the pre-trained GNN. Starting from the current state node $\vec{h}_s$, we enumerate all possible actions $\vec{a}_i \in |\mathcal{A}|$ to connect neighbors via $\vec{h}(s) \xrightarrow{\vec{a}_i} \vec{h}(s_i')$. As a result, each node has degree $|\mathcal{A}|$, and graph size grows even faster as it expands deeper. We propose to use a neighbor sampling method so that we only expand a small subset of actions. However, the important question is which actions to select. Value iteration algorithm is a DP algorithm whose update rule is the Bellman optimality equation (Eq.1). The max aggregator iterates through the entire action space $a \in \mathcal{A}$ to ensure that we get the optimal solution at each iteration. Therefore, the graph constructed should allow our pre-trained GNN to predict optimally at each layer. It is thus critical that we can include the action that produces a good approximation of the state-value function in our sampling.

Below, we propose four possible methods to sample $K$ actions from $\mathcal{A}$, where $K \ll |\mathcal{A}|$ is a fixed number, under the context of value-iteration-based planning.

### 4.5.1 Gaussian methods

Gaussian distribution is a common baseline policy distribution for continuous action spaces, and it is straightforward to interpret. Furthermore, it discourages extreme actions while encouraging neutral ones with some level of continuity, which suits the requirement of many planning problems. We propose two variants of sampling policy based on Gaussian distribution.

**(a) Manual-Gaussian**: A Gaussian distribution is used to randomly sample action values in each dimension $a_i \in \mathcal{A}_i$, which are stacked together as a final action vector $\vec{a} = [a_0, ..., a_{D-1}]^T \in \mathcal{A}$. We repeat for $K$ times to sample a subset of $K$ action vectors. We set the mean $\mu = N/2$ and standard deviation $\sigma = N/4$, where $N$ is the number of discrete action bins. These two parameters are chosen to spread a reasonable distribution over $[0, N-1]$. Outliers and non-integers are rounded to the nearest whole number within the range of $[0, N-1]$.

**(b) Learned-Gaussian**: The two parameters manually chosen in the previous method pose a constraint on placing the median action in each dimension as the most likely. Here instead, two fully-connected linear layers are used to separately estimate the mean $\mu$ and standard deviation $\sigma$. They take the state embedding $\vec{h}_s$ from Encoder and output parameter estimations for each dimension. We use the reparameterization trick [30] to make the sampling differentiable.

### 4.5.2 Parameter reuse

Gaussian methods still restrain a fixed distribution on the sampling distribution, which may not necessarily fit. Previous work [23] studied the action sampling problem on policy evaluation and improvement. They reasoned that since the actions selected by the policy are expected to be more valuable, we can directly use the policy for sampling.

**(c) Reuse-Policy**: We can reuse Policy layer $P(\vec{h}_s, \vec{\mathcal{X}}_s)$ to sample the actions when we expand the graph in Executor. This is equivalent to using the policy distribution $\pi^* = p(\vec{a}|s)$ as the neighbor sampling distribution. However, the second input $\vec{\mathcal{X}}_s$ for Policy layer comes from Executor, which is not available at the time of constructing the graph. It is filled up by setting $\vec{\mathcal{X}}_s = \vec{0}$ as placeholders.

### 4.5.3 Learn to expand

Lastly, we can also use a separate layer to learn the neighbor sampling distribution.

**(d) Learned-Sampling**: This uses a fully-connected linear layer that consumes $\vec{h}_s$ and produces an output dimension of $|N \cdot D|$. It is expected to learn the optimal neighbor sampling distribution in a factorized joint manner, same as Figure 2(a). The outputs are logits for $D$ categorical distributions, where we used Gumbel-Softmax [31] for differentiable sampling actions in each dimension, together producing $\vec{a} = [a_1, ..., a_D]^T$.

**Table 1:** Summary of the four neighbor sampling policies on their pros & cons.

| Manual-Gaussian | (+) Sample-efficient as no training is required. | (-) Gaussian distribution may not fit. (-) Assume the median as the most likely. |
|---|---|---|
| Learned-Gaussian | (+) More flexible choice of distribution range. | (-) Gaussian distribution may not fit. (-) More parameters requires more training. |
| Reuse-Policy | (+) Parameter reuse. (+) Policy distribution alignment. | (-) Misalignment in input format due to the unavailability of $\vec{\mathcal{X}}_s$ in Executor. |
| Learned-Sampling | (+) Dedicated distribution learning. | (-) More parameters requires more training. |

## 5 Results

### 5.1 Classic Control

To evaluate the performance of CNAP agents, we first ran the experiments on a relatively simple MountainCarContinuous-v0 environment from OpenAI Gym Classic Control suite [32], where the action space was one-dimensional. The training of the agent used PPO under 20 rollouts with 5 training episodes each, so the training consumed 100 episodes in total.

We compared two variants of CNAP agents: **"CNAP-B"** had its Executor pre-trained on a type of binary graph that aimed to simulate the bi-directional control of the car, and **"CNAP-R"** had its Executor pre-trained on random synthetic Erdős-Rényi graphs. In Table 2, we compared both CNAP agents against a **"PPO Baseline"** agent that consisted of only the Encoder and Policy/Tail modules.

Both the CNAP agents outperformed the baseline agent for this environment, indicating the success of extending XLVIN onto continuous settings via binning.

**Table 2:** Mean rewards for MountainCarContinuous-v0 using PPO Baseline and two variants of CNAP agents. All three agents ran on 10 action bins, and were trained on 100 episodes in total. Both CNAP agents executed one step of value iteration. The reward was averaged over 100 episodes and 10 seeds.

| Model | MountainCarContinuous-v0 |
|---|---|
| PPO Baseline | -4.96 $\pm$ 1.24 |
| CNAP-B | 55.73 $\pm$ 45.10 |
| CNAP-R | **63.41** $\pm$ 37.89 |

### 5.1.1   Effect of GNN width and depth

We then studied the effects of CNAP agents' two hyperparameters. In Table 3, we varied the number of action bins into which the continuous action space was discretized. The results showed that 10 action bins led to the best performance, suggesting the importance of balancing how much information we can sacrifice for discretization. On the other hand, a larger number of action bins results in a larger graph size, requiring more samples to train, hindering sample efficiency. We provide additional results on increasing the number of bins to 50 and 100 in Appendix A.1 which led to even worse results.

In Table 4, we varied the number of GNN steps, corresponding to the number of steps we simulated in the value iteration algorithm. A degradation in performance is also observed, with 1 GNN step bringing the best performance. One possible reason was also how the number of training samples might also not be sufficient when given larger graph depths. Also, a deeper graph required repeatedly applying the Transition module, where the imprecision might add on, leading to inappropriate state embeddings and hence less desirable results.

More ablation results on the combined effect of varying both the width and depth on CNAP-R can be found in Appendix A.2.

**Table 3:** Mean rewards for MountainCarContinuous-v0 using Baseline and CNAP agents by varying number of action bins, i.e., width of graph. The results were averaged over 100 episodes and 10 seeds.

| Model | Action Bins | MountainCar-Continuous |
|---|---|---|
| PPO | 5 | -2.16 $\pm$ 1.25 |
| | 10 | -4.96 $\pm$ 1.24 |
| | 15 | -3.95 $\pm$ 0.77 |
| CNAP-B | 5 | 29.46 $\pm$ 57.57 |
| | 10 | 55.73 $\pm$ 45.10 |
| | 15 | 22.79 $\pm$ 41.24 |
| CNAP-R | 5 | 20.32 $\pm$ 53.13 |
| | 10 | **63.41** $\pm$ 37.89 |
| | 15 | 26.21 $\pm$ 46.44 |

**Table 4:** Mean rewards for MountainCarContinuous-v0 using CNAP agents by varying number of GNN steps, i.e., depth of graph. The results were averaged over 100 episodes and 10 seeds.

| Model | GNN Steps | MountainCar-Continuous |
|---|---|---|
| CNAP-B | 1 | 55.73 $\pm$ 45.10 |
| | 2 | 46.93 $\pm$ 44.13 |
| | 3 | 40.58 $\pm$ 48.20 |
| CNAP-R | 1 | **63.41** $\pm$ 37.89 |
| | 2 | 34.49 $\pm$ 47.77 |
| | 3 | 43.61 $\pm$ 46.16 |

## 5.2 MuJoCo

We then ran experiments on more complex environments from OpenAI Gym's MuJoCo suite [19, 32] to evaluate how CNAPs could handle the high increase in scale. Unlike the Classic Control suite, the MuJoCo environments have higher dimensions in both its observation and action spaces. We started by evaluating CNAP agents in two environments with relatively lower action dimensions, and then we moved on to two more environments with much higher dimensions. The discretization of the continuous action space also implied a combinatorial explosion in the action space, resulting in a large graph constructed for the GNN. We used the proposed factorized joint policy from Section 4.4 and the neighbor sampling methods from Section 4.5 to address the limitations.

### 5.2.1 On low-dimensional environments

In Figure 3, we experimented with the four sampling methods discussed in Section 4.5 on Swimmer-v2 (action space dimension of 2) and HalfCheetah-v2 (action space dimension of 6). We chose to take the number of action bins $N = 11$ for all the experiments following [22], where the best performance on MuJoCo environments was obtained when $7 \leq N \leq 15$. The number of neighbours to expand was set to $K = 10$, so that we could evaluate the four neighbour expansion policies when sampling a very small subset of actions. In all cases, CNAP outperformed the baseline in the final performances. Moreover, Manual-Gaussian and Reuse-Policy were the most promising sampling strategies as they also demonstrated faster learning, hence better sample efficiency. This pointed to the benefits of parameter reuse and the synergistic improvement between learning to act and learning to sample relevant neighbors, as well as the power of a well-chosen manual distribution. We also note that choosing a manual distribution can become non-trivial when the task becomes more complex, especially if choosing the average values for each dimension is not the most desirable. Our work acts as a proof-of-concept of sampling strategies and leaves the choice of parameters for future studies.

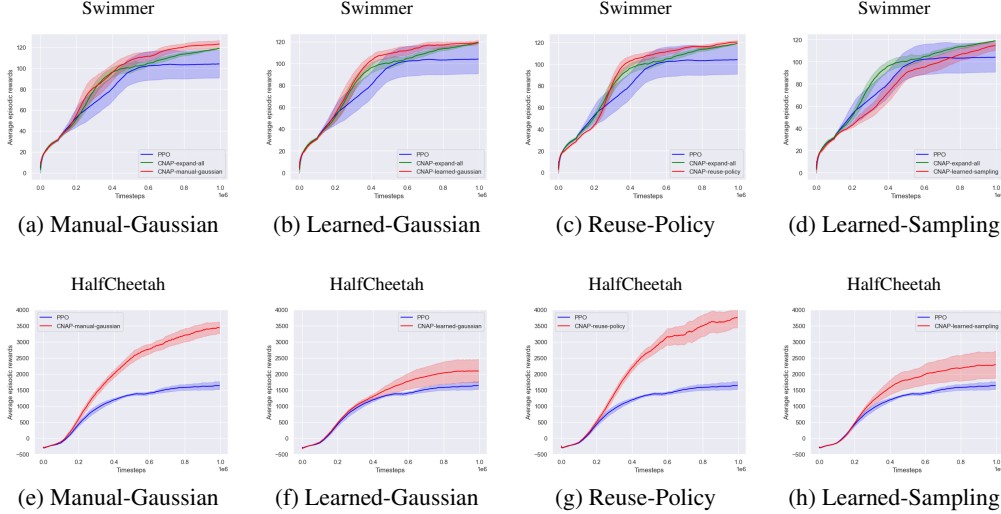

(a) Manual-Gaussian    (b) Learned-Gaussian    (c) Reuse-Policy    (d) Learned-Sampling

(e) Manual-Gaussian    (f) Learned-Gaussian    (g) Reuse-Policy    (h) Learned-Sampling

**Figure 3:** Average rewards over time for CNAP (red) and PPO baseline (blue), in Swimmer (action dimension=2) and Halfcheetah (action dimension=6), using different sampling methods. In Swimmer, CNAP with sampling methods were compared with the original version by expanding all actions (green). In (a)(e), the actions were sampled using Gaussian distribution with mean=$N/2$ and std=$N/4$, where $N$ was the number of action bins used to discretize the continuous action space. In (b)(f), two linear layers were used to learn the mean and std, respectively. In (c)(g), the Policy layer was reused in sampling actions to expand. In (d)(h), a separate linear layer was used to learn the optimal neighbor sampling distribution. The mean rewards were averaged over 100 episodes, and the learning curve was aggregated from 5 seeds.

### 5.2.2 On high-dimensional environments

We then further evaluated the scalability of CNAP agents in more complex environments where the dimensionality of the action space was significantly larger, while retaining a relatively low-data

regime ($10^6$ actor steps). In Figure 4, we compared all the previously proposed CNAP methods on two environments with highly complex dynamics, both having an action space dimension of 17. In the Humanoid task, all variants of CNAPs outperformed PPO, acquiring knowledge significantly faster.

Particularly, we found that *nonparametric* approaches to sampling the graph in CNAP (e.g. manual Gaussian and policy reuse) acquired this knowledge significantly faster than any other CNAP approach tested. This supplements our previous results well, and further testifies to the improved learning stability when the sampling process does not contain additional parameters to optimise.

We also evaluated all of the methods considered against PPO on the HumanoidStandup task, with all methods learning to sit up, and no apparent distinction in the rate of acquisition. However, we provide some qualitative evidence that the solution found by CNAP appears to be more robust in the way this knowledge acquired—see Appendix A.3.

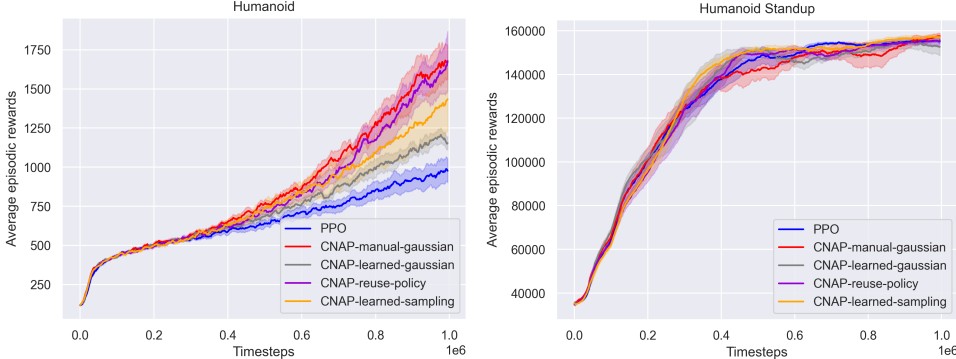

**Figure 4:** Average rewards over time for CNAP (red) and PPO baseline (blue), in Humanoid (action dimension=17) and HumanoidStandup (action dimension=17), using the four sampling methods.

### 5.2.3 Qualitative interpretation

We also captured the video recordings of the interactions between the agents and the environments to provide a qualitative interpretation to the results above. We chose to look at the selected frames (Appendix A.3) at equal time intervals from one episode after the last training iteration by CNAP (Manual-Gaussian) and PPO Baseline, respectively.

In HalfCheetah task, the agent instructed by PPO Baseline fell over quickly and never managed to turn it back. However, CNAP's agent could balance well and kept running forward. This observation could support the higher average episodic rewards gained by CNAP agents than by PPO Baseline in Figure 3. Similarly for Humanoid task, PPO Baseline's humanoid stayed stationary and lost balance quickly, while CNAP's humanoid could walk forward in small steps. This observation aligned with the results in Figure 4 where the gain from CNAP was significant. In addition, we note that, although quantitatively CNAP agent did not differentiate from PPO Baseline in HumanoidStandup task as shown in Figure 4, for the trajectories we observed, it successfully remained in a sitting position, while the PPO Baseline fell quickly.

## 6 Conclusion

We present CNAP, a method that generalizes implicit planners to continuous action spaces for the first time. In particular, we study implicit planners based on neural algorithmic reasoners and the unstudied implications of not having precise alignment between the learned graph algorithm and the setup where the executor is applied. To deal with the challenges in building the planning tree, as a result of the continuous, high-dimensional nature of the action space, we combine previous advancements in XLVIN with binning, as well as parametric and non-parametric neighbor sampling strategies. We evaluate the agent against its model-free variant, observing its efficiency in low-data settings and consistently better performance than the baseline. Moreover, this paves the way for extending other implicit planners to continuous action spaces and studying neural algorithmic reasoning beyond strict applications of graph algorithms.

## Acknowledgements

We would like to thank Adrià Puigdomènech Badia, Doina Precup, and all anonymous reviewers from LoG Conference 2022 and GroundedML Workshop at ICLR 2022 for their detailed and constructive feedback, which helped to greatly strengthen this paper. Yu He would also like to thank Han Xuanyuan for reading an earlier version of this manuscript and for his support throughout.

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

## A Appendix

### A.1 Even larger number of action bins

In Table 5, we increase the number of action bins to even larger numbers of 50 and 100, where a further degradation is observed. More action bins results in a larger graph that requires more training samples to be consumed, which compromises the sample efficiency.

**Table 5:** Mean rewards for MountainCarContinuous-v0 using CNAP-R with 1 GNN step by varying number of action bins (width of graph). The results were averaged over 100 episodes and 10 seeds.

| Number of action bins | MountainCar-Continuous |
|---|---|
| 10 | **63.41** $\pm$ 37.89 |
| 50 | -8.88 $\pm$ 1.13 |
| 100 | -13.50 $\pm$ 1.60 |

## A.2 Combined effect of varying width and depth of GNN

We show the combined effects of varying the number of GNN steps and action bins of the graph in Table 6. We observe that within each row, an appropriate number of action bins such as 10 obtains sufficient information from discretization. Within each column, a smaller GNN step of 1 is generally more preferrable.

**Table 6:** Mean rewards for MountainCarContinuous-v0 using CNAP-R by varying number of GNN steps (depth of graph), and number of action bins (width of graph). The results were averaged over 100 episodes and 10 seeds.

| | Number of action bins | | |
|---|---|---|---|
| GNN Steps | 5 | 10 | 15 |
| 1 | 20.32$\pm$53.13 | **63.41**$\pm$37.89 | 26.21$\pm$46.44 |
| 2 | 25.33$\pm$47.08 | 34.49$\pm$47.77 | 17.15$\pm$46.26 |
| 3 | 19.23$\pm$54.19 | 43.61$\pm$46.16 | 18.99$\pm$44.69 |

## A.3 Selected frames for MuJoCo tasks

Swimmer: PPO

Swimmer: CNAP

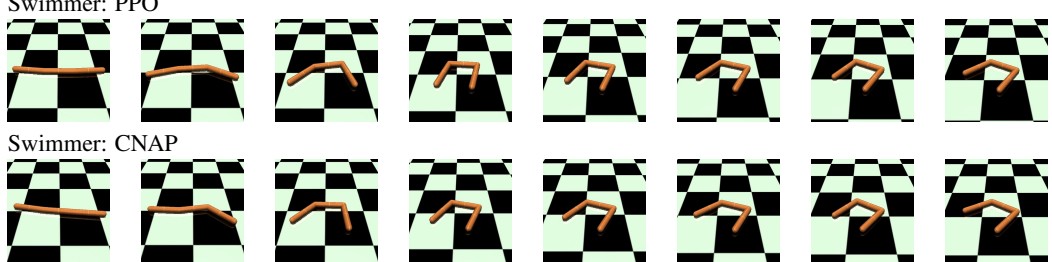

**Figure 5:** Selected frames of two agents in Swimmer

As seen in Figure 5, CNAP could fold itself slightly faster than PPO Baseline in this episode and swam more quickly.

HalfCheetah: PPO

HalfCheetah: CNAP

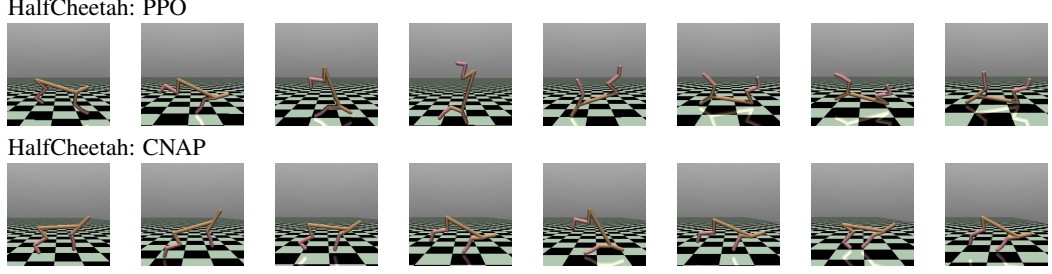

**Figure 6:** Selected frames of two agents in HalfCheetah

From Figure 6's HalfCheetah task, we can see the agent instructed by PPO Baseline fell over quickly and never managed to turn it back. However, CNAP's agent could balance well and kept running

forward. This observation could support the higher average episodic rewards gained by CNAP agents than by PPO Baseline in Figure 3.

Humanoid: PPO

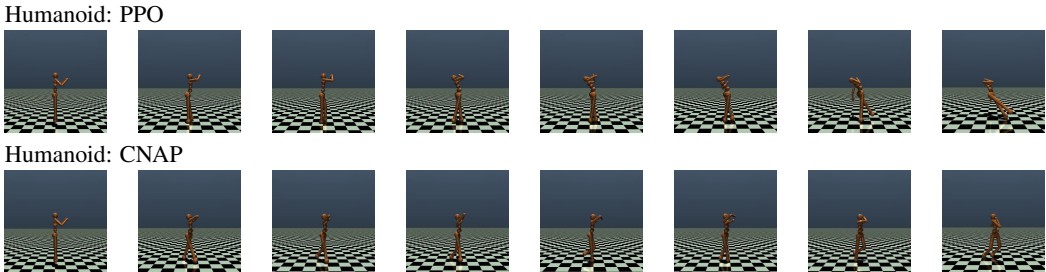

Humanoid: CNAP

**Figure 7:** Selected frames of two agents in Humanoid

Similarly, in Figure 7's Humanoid task, PPO Baseline's humanoid stayed stationary and lost balance quickly, while CNAP's humanoid could walk forward in small steps. This observation aligned with the results in Figure 4 where the gain from CNAP was significant.

HumanoidStandup: PPO

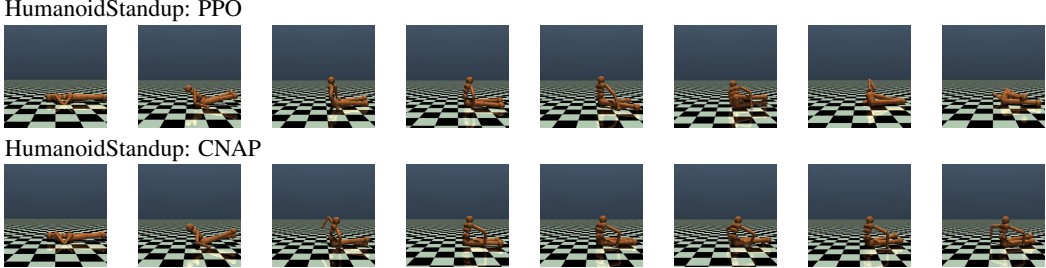

HumanoidStandup: CNAP

**Figure 8:** Selected frames of two agents in HumanoidStandup

Then we noticed that although in HumanoidStandup task, the quantitative performances between PPO Baseline and CNAP were similar, Figure 8 revealed some different results. Both agents did not manage to stand up, explaining why the episodic rewards were similar numerically. However, the PPO Baseline agent lost balance and fell back to the ground while the CNAP agent remained sitting, trying to get up. Therefore, the CNAP qualitatively performed better in this example.

