# OpenReview forum: "Continuous Neural Algorithmic Planners"
_logconference.io/LOG/2022/Conference — LoG 2022 Poster_

### Official Review · Reviewer_VqqS · 2022-10-15

**Overall Score:** 6
**Confidence:** 2

**Review:**

# Summary:
The author mainly extended the discrete version of the XLVIN model to a continuous one and applied it to continuous tasks like mountainCar, MuJoCo scenes, etc.

With an appropriate neighbor sampling method, the proposed algorithm model helps to improve the efficiency of the graph constructing process. CNAP is Motivating a wider range of applications that utilize the graph algorithms and bring benefits through implicit planning.

## Pros and Cons:
### Strength

- The proposed CNAP overcomes the previous version’s discrete limitation by discretization in action space. Making the model more transferable to other realistic scenario tasks.

- The author proposed clear 4 solutions to tackling the action-selection challenge and did ablation studies on each method to justify the effectiveness.

### Weakness

- The author might consider expressing the first challenge which lies in the bottleneck of the “ Policy module that produces the action probabilities” in a complete logic flow, to strengthen “the problem in this module”, “why to choose Factorized joint policy”, “how it can help to solve the challenge”, and “the effectiveness of such an attempt”. Otherwise, it can be hard to capture the innovation here.

- And there could be a few aspects remaining unclear, more explanation should be made, seen Questions of the paper.

- The author mentioned a challenge in the Model Architecture section which might hinder the quick understanding, of this aspect, a suggestion to propose the main challenges at the very beginning of the article, so that readers would keep in mind what is the key point of the work.

## Support of Recommendation:

I suggest Weak reject based on previous pros and cons.

## Questions to clarify in the paper:

- In section2.4, how could it be seen that there exists an alignment between GNN and DP from mapping nodes representation h(s) to Answer[k][i]? Detailed explanations are expected.

- What does ‘good sample complexity’ mean? Is ‘Good’ here mean lower complexity?

- What does the [v1, v2] in the 4.2 respectively refer to? It may lead to an unclear understanding in the following context of a(i)k’s representation. A more explicit statement should be provided.

- There is a misplacement of a hat on the p in 4.3, line 2. The same problem happened in the next few contexts as well, the author might examine and modify them in detail.

- How does the Factorized joint policy solve the problem? The author mainly talked about the consequence of using such a method, which is: the exponential explosion of the action bin turned linear. However, since this is one of the main solutions to the paper’s first challenge, so emphasis could be focused on this part.

---

### Official Review · Reviewer_dYGJ · 2022-10-17

**Overall Score:** 6
**Confidence:** 2

**Review:**

This paper proposes a modification of XLVIN for reinforcement learning with continuous action spaces. To this end, the authors propose (1) action discretization, (2) factorized joint policy, and (3) various neighborhood sampling techniques to limit the expansion of the graph.

Strength: This paper proposes a solid way to generalize XLVIN to continuous action space. All the proposed method seems reasonable.

Weakness: While the proposed methods are solid, they lack novelty. Discretization of action space and factorized joint policy has been proposed by [1]. Learning the neighborhood sampling distribution (rollout policy) has been investigated by [2]. In the original XLVIN paper, using [1,2] to expand XLVIN is explicitly mentioned as its future work. Without methodological novelty, I am concerned about the significance of this work.

The proposed algorithm is only compared to PPO. Additional baselines like soft actor-critic would be useful to improve empirical validation of the paper.

Overall, I recommend weak rejection for the weaknesses.

---

### Official Review · Reviewer_2YhM · 2022-10-19

**Overall Score:** 6
**Confidence:** 2

**Review:**

Summary:
This paper proposes a method to value iteration algorithm in reinforcement learning. Based on eXecuted Latent Value Iteration Net (XLVIN), it further proposes i) a factorized joint policy to solve the problem of combinatorial explosion in action space; ii) neighbor sampling methods to narrow the action searching space. Experiment including qualitative results and quantitive results shows the proposed method’s effectiveness.

Strength:
1. This paper proposes a more generalized algorithmic planner on continuous action spaces, which suits more application problems.
2. Both the factorized joint policy and the neighbor sampling methods are important for efficient action searching in value iteration.
3. The experiment result is sufficient to show the effectiveness of the proposed method.

Weakness:
1. In section 4.2, the challenge cannot be reflected in the problem formulations. Assuming the action space as a multi-dimensional space is general and better for understanding.
2. In section 4.4.1’s Manual-Gaussian, the method repeats for K times to sample a subset of K action vectors. In every sampling, is the sampling result a whole action vector [a0, ..., aD-1] or a single action ai? If it is the former, how can a gaussian distribution model a multi-dimensional distribution, maybe it should be multi-variate gaussian? If it is later, the sampling should be repeated for K^D times which is exploded. This part is not clear enough and more detail should be stated.
3. If each dimension has a different number of actions in section 4.3, what will the factorized joint policy be? Can it still work?
4. In Fig.4 in the experiment part, why does the manual gaussian achieve the best result? What is the disadvantage of learning-based sampling methods?
5. It is better to analyze the advantages and disadvantages of four different neighboring sampling method in the experiment part.

---

### Official Review · Reviewer_ZTDw · 2022-10-22

**Overall Score:** 6
**Confidence:** 4

**Review:**

Summary. This paper proposes a framework namely CNAP to solve algorithmic reasoning in continuous planning using GNNs. CNAP extends XLVIN to deal with the continuous action space through discretization. It applies factorized joint policy to reduce the exponential policy space to a linear space. It further adopts four neighborhood sampling methods to select a subset of actions to expand, instead of walking through all possible actions. Through comprehensive experiments, the paper shows the effectiveness of CNAP for continuous planning and outperforms the baseline method such as PPO.

Recommendation. I recommend this paper as “Weak Accept”, considering that the paper studies a novel problem of continuous planning using GNNs, but the technical contribution is less significant. See detailed evaluation below.

[Strengths]

S1. The paper proposes a framework namely CNAP to solve algorithmic reasoning in continuous reinforcement learning, which is a novel and interesting problem.

S2. Comprehensive experiments are conducted for CNAP variants to compare to baselines, both quantitatively and qualitatively, and the results are clearly explained.

S3. The paper is well-written with related work properly cited and it is easy to follow with good readability.

[Need to improve]

N1. The major technical contributions are less significant. See D1.

N2. More in-depth analysis or experiments can be conducted. See D2 and D3.

[Details]

D1. CNAP extends XLVIN [17] to deal with continuous action space by discretization. The solution is a direct application of existing work such as factorized joint policy [22] for reducing policy space, reuse-policy [23] and learned-gaussian [30] for sampling, which limits the technical novelty.

D2. CNAP uses a fixed number K for action sampling, but the paper does not discuss how this affects the performance. In addition, for message passing GNNs, the paper does not study how specific GNN models affect the results.

D3. Section 5.1.1, the paper mentions “The two agents performed best with 10 action bins and one GNN step.” But both Table 2 and Table 3 only vary a single parameter, which cannot show this conclusion. A table that combines the two dimensions (width, depth) will be better.

Also, there are too few discretized actions when there are 5, 10, and 15 bins, which may not be good for studying continuous planning. Perhaps, varying number of bins exponentially (e.g., 10, 100, 1000) will show clearer trends.

Minors.

D4. Section 4.2, the symbol “a_i^k” stands for action bins, which seems to overload symbol “a” for actions. Change a_i^k to something else should be preferred, e.g., b_i^k.

D5. Figure 3. The two rows of figures correspond to two benchmarks for Swimmer and Halfcheetah, respectively, but there are no labels to show them, and both have subfigures (a), (b), (c), and (d), which can be confusing.

D6. Figure 4. In caption, it only says “using Manual-Gaussian and Reuse-Policy sampling methods.”, but there are also results of learned-gaussian and learned-sampling shown in the figure.

D7. In Section 4.4, two kinds of numbering are mixed. Both subsubsections (4.4.1 to 4.4.3) and paratheses around numbers before each method ((1), (2), (3), and (4)) are in use.

D8. In Figure 3, why is there no line for CNAP-expand-all in the second row, just like what it does in the first row? The format looks inconsistent here.

For rebuttal, please address above D1-D8.

---

### Meta-Review · Area_Chair_57Pq · 2022-11-14

**Confidence:** 4
**Recommendation:** Accept

**Meta Review:**

# Summary

This paper builds upon the recently introduced XLVIN neural algorithmic planner. Briefly, this architecture leverages a graph neural network that simulates value iteration algorithm in deep reinforcement learning agent. However, it is limited to discrete action spaces. Based on this context, this paper proposed to extend XLVIN to continuous action spaces by discretization. Experiments are carried out on standard continuous control tasks (e.g., MuJoCo) and shows that the architecture brings gains in low-data settings and outperforms model-free baselines.

# Recommendation

This paper was well received by the reviewing team. We have a broad agreement towards an acceptance and the author response was helpful to converge towards this agreement. The concerns and questions were successfully addressed by the authors. Here are the three main points motivating this recommendation:

1. The idea to solve algorithmic reasoning in continuous space is novel and interesting. This enables the model to be used for other realistic tasks.

2. The experiments are convincing. Comparisons are conducted against different baselines, both quantitatively and qualitatively. Besides, ablation studies are proposed and the results are clearly explained. As a small concern, it would have been great to have a comparison with SAC as well.

3. Although the idea to solve algorithmic reasoning in continuous space is novel and interesting, the technical contribution is less significant (see the detailed discussion with Revierwer ZTDw). This explains why the recommendation score is not higher.

# Conclusion

To conclude, the reviewing team that the paper deserves to be accepted (no spotlight). Besides, we strongly encourage the authors to incorporate the suggestions raised in the final version of the paper.

---

### Decision · Program_Chairs · 2022-11-23

Accept (Poster)